# Optimization of Multimodal Nanoparticles Internalization Process in Mesenchymal Stem Cells for Cell Therapy Studies

**DOI:** 10.3390/pharmaceutics14061249

**Published:** 2022-06-12

**Authors:** Mariana P. Nucci, Javier B. Mamani, Fernando A. Oliveira, Igor S. Filgueiras, Arielly H. Alves, Matheus H. Theinel, Luiz D. Rodrigues, Luciana Marti, Lionel F. Gamarra

**Affiliations:** 1Hospital Israelita Albert Einstein, São Paulo 05652-000, Brazil; mariana.nucci@hc.fm.usp.br (M.P.N.); jbusta25@gmail.com (J.B.M.); fernando.ao@einstein.br (F.A.O.); igorsfilgueiras@hotmail.com (I.S.F.); arielly.alves@einstein.br (A.H.A.); nutrimatheusht@gmail.com (M.H.T.); luciana.marti@einstein.br (L.M.); 2LIM44—Hospital das Clínicas da Faculdade Medicina da Universidade de São Paulo, São Paulo 05403-000, Brazil; 3Faculdade de Medicina Veterinária e Zootecnia da Universidade de São Paulo, São Paulo 05508-270, Brazil; ld.rodriguesvet@gmail.com

**Keywords:** multimodal nanoparticle, labeling, magnetic field, incubation time, mesenchymal stem cell, cell therapy, stroke, near-infrared fluorescence image, bioluminescence, ICP-MS

## Abstract

Considering there are several difficulties and limitations in labeling stem cells using multifunctional nanoparticles (MFNP), the purpose of this study was to determine the optimal conditions for labeling human bone marrow mesenchymal stem cells (hBM-MSC), aiming to monitor these cells in vivo. Thus, this study provides information on hBM-MSC direct labeling using multimodal nanoparticles in terms of concentration, magnetic field, and period of incubation while maintaining these cells’ viability and the homing ability for in vivo experiments. The cell labeling process was assessed using 10, 30, and 50 µg Fe/mL of MFNP, with periods of incubation ranging from 4 to 24 h, with or without a magnetic field, using optical microscopy, near-infrared fluorescence (NIRF), and inductively coupled plasma mass spectrometry (ICP-MS). After the determination of optimal labeling conditions, these cells were applied in vivo 24 h after stroke induction, intending to evaluate cell homing and improve NIRF signal detection. In the presence of a magnetic field and utilizing the maximal concentration of MFNP during cell labeling, the iron load assessed by NIRF and ICP-MS was four times higher than what was achieved before. In addition, considering cell viability higher than 98%, the recommended incubation time was 9 h, which corresponded to a 25.4 *p*g Fe/cell iron load (86% of the iron load internalized in 24 h). The optimization of cellular labeling for application in the in vivo study promoted an increase in the NIRF signal by 215% at 1 h and 201% at 7 h due to the use of a magnetized field during the cellular labeling process. In the case of BLI, the signal does not depend on cell labeling showing no significant differences between unlabeled or labeled cells (with or without a magnetic field). Therefore, the in vitro cellular optimized labeling process using magnetic fields resulted in a shorter period of incubation with efficient iron load internalization using higher MFNP concentration (50 μgFe/mL), leading to significant improvement in cell detection by NIRF technique without compromising cellular viability in the stroke model.

## 1. Introduction

Ischemic stroke and other neurological disorders can cause brain damage due to neuron and glial cell loss, resulting in significant disability and ranking among the major reasons of mortality [1,2]. Advanced cellular therapy has been described as an alternative or as an important adjuvant in medical therapy due to its potential for tissue repair after a nervous system injury. A major cause of brain damage is ischemic stroke, which is the second leading reason of death worldwide and the third reason for disability in the world (WHO, 2021) [3].

A recent review in stroke clinical trials (mainly phases I or II) demonstrated that stem cells are safe, and now there is an ongoing evaluation in neuro-regeneration and sequelae reduction. However, the heterogeneity of these studies’ design and evaluation criteria can interfere with clear conclusions about the efficacy of these treatments [4,5,6,7,8].

Mesenchymal stem cells (MSCs) have been used as a treatment for stroke due to their ability to promote immunomodulation, angiogenesis, and support neurogenesis in the injured area [9]. A number of clinical [10,11,12] and pre-clinical [13,14,15] studies have demonstrated certain benefits of this therapy. However, some studies reported lower translational efficacy of MSCs as a therapy for Stroke [16]. One of the most difficult aspects of understanding several studies’ results is the confirmation that if administered, cells were able to reach the lesion site and, once there, if they were viable enough to support modulation and neurological recovery [17].

One alternative for monitoring MSCs after transference to an injured brain is the use of molecular imaging techniques. These techniques allow non-invasive longitudinal cell tracking and quantitative data gatherings, such as magnetic resonance imaging (MRI), nuclear imaging, optical imaging such as fluorescence and bioluminescence imaging, and multimodal molecular imaging, which combines two or more imaging methods [18,19]. The selection of molecular imaging technology is based on a balance between advantages and disadvantages, taking into account sensitivity, resolution, image depth, safety, and labeling process, among other parameters. For example, MRI has an unlimited penetration capacity but lacks sensitivity, demanding larger numbers of cells in a specific region to be detected by a contrast agent [20,21]. Nuclear imaging techniques offer an unlimited penetration capacity, good sensitivity, and spatial resolution but require the use of radioactive agents, which have a limited half-life depending on the radiotracer. Optical imaging techniques, on the other hand, offer a higher spatial resolution and sensitivity than the previous ones but have a limited penetration capacity and are still not available for clinical practice [19,21,22].

Multimodal molecular imaging integrates multiple types of imaging modalities that simultaneously give information on physiological and anatomical features of the disease [23,24], compensating for the shortcomings and complementing the advantages of each molecular imaging technique.

Every molecular imaging approach requires the MSCs to be labeled with a specific agent for further tracking in in vivo models [18,22]. One of the strategies for monitoring MSCs is their labeling using nanomaterials, such as magnetic nanoparticles, fluorescent nanoparticles, and other types of nanoparticles, such as carbon nanotubes or nanoparticulate compounds. However, these cells should be labeled with an adequate quantity of nanoparticles for their detection without causing cellular toxicity [25,26,27,28,29]. Despite promising results in cell tracking using nanomaterials, some challenges remain to be solved, such as a lack of sensitivity in molecular imaging techniques, adequate physical-chemical properties of nanoparticles for their internalization process, and optimal concentration in labeled cells, image acquisition parameters, and so on. To overcome these limitations, composite nanoparticles were designed to serve as a contrast agent for multimodal imaging, transcending the limits of each individual imaging technique [30,31,32].

There are certain actions that can improve nanoparticle internalization in cells, such as increasing the period of incubation, surface modification, transfection agents (Poly-D-Lysine and protamine sulfate), and application of magnetic field already evaluated in previous studies [33,34,35,36,37]. The magnetic field attracts the nanoparticle, which contributes to cellular internalization throughout the labeling process [38]. Newton’s law and electromagnetic theory govern these forces in the nanoparticles, taking diffusion, convection, and the magnet field into account [38,39]. The magnetic force that attracts the nanoparticles towards the cells in culture is responsible for the highest efficiency of the cell labeling procedure [38]. Aside from having appropriate physicochemical attributes that do not cause toxicity [40].

Another method for labeling MSCs is to genetically modify them to express fluorescent or bioluminescent agents such luciferase, which can generate light when combined with its substrate luciferin under optimum conditions [41,42]. In addition to tracking, the bioluminescence image can be used to assess the viability of MSCs in the stroke region, as the cells will only release light if they are alive [41].

The goal of this study, considering the challenges and limitations associated with cell labeling, was to evaluate the best conditions to optimize a method for direct labeling human bone marrow-derived mesenchymal stem cell (hBM-MSC) using multimodal nanoparticles and determining the best concentration, magnetic field and period of incubation to maintain adequate cell viability and good cellular homing signal for in vivo studies.

## 2. Materials and Methods

### 2.1. Obtainment of Human Bone Marrow Mesenchymal Stem Cells (hBM-MSCs)

The Hospital Israelita Albert Einstein (HIAE, São Paulo, Brazil) Ethics Research Committee approved this study under CAAE number: 64288917.0.0000.0071, and informed consent was received from all participants included in this study. The mononuclear cells were obtained from the bone marrow of 10 healthy donors. After bone marrow aspiration, a sample was diluted with phosphate-buffered saline (PBS) (Gibco^®^, Carlsbad, CA, USA), and mononuclear cells were separated by density gradient using Ficoll-Paque™ Premium (1.084 g/mL) (GE Healthcare, Uppsala, Sweden). Briefly, cells were centrifuged (without brake) for 30 min, at 500× *g* and 21 °C. Then, mononuclear cells were removed from the interface, transferred to another tube and washed three times with PSB. The cells were then resuspended in Dulbecco’s Modified Eagle’s Medium/Nutrient Mixture F-12 (DMEM/F-12) (GIBCO^®^ Invitrogen Corporation, Carlsbad, CA, USA) supplemented with fetal bovine serum (FBS) (GIBCO^®^ Invitrogen Corporation, Carlsbad, CA, USA), and 1% antibiotic-antimycotic solution (GIBCO^®^ Invitrogen Corporation, Carlsbad, CA, USA). Next, the cells in suspension were transferred to a T25 cell culture flask and placed in a 5% CO_2_ incubator at 37 °C. The cells in the culture were evaluated every day, and the supplemented culture media was replaced every other day. Cells were maintained in these culture conditions until they reached confluence, and TrypLE™ Express Enzyme (GIBCO^®^ Invitrogen Corporation, Carlsbad, CA, USA) was used to detach and pass the cells as needed.

### 2.2. hBM-MSC Immunophenotypic Characterization

The immunophenotyping of hBM-MSC unlabeled and labeled with 50 µg Fe/mL of MFNP during 24 h (the maximum incubation time analyzed in this study) was performed using the following antibodies: CD19-FITC (clone:4G7) and CD34-PE (clone: 8G12) from BD Biosciences, San Jose, CA, USA; CD14-Alexa 700 (clone: M5E2), CD29-APC (clone: MAR04), CD35-FITC (clone: E11), CD44-PerCP-Cy5.5 (clone: G44-26), CD73-PE (clone: AD2), CD90-PE-Cy7 (clone: 5E10), HLA-DR-APC-H7 (clone: G46-6), CD106-FITC (clone: 51-10C9) all from BD Pharmingen, San Diego, CA, USA; CD31-V450 (clone: WM59), CD45-V500 (clone: H130), CD105-PE-CF546 (clone: 266) all from BD Horizon, San Jose, CA, USA, unstained samples and fluorescence minus one (FMO) was used as a fluorescence background control. Briefly, 1 × 10^5^ cells were incubated in the dark/room temperature for 30 min in the presence of antibodies at concentrations recommended by the manufacturers. Cells were then washed with a buffered solution, and at least 10,000 events were acquired using FACS LSRII FORTESSA equipment (BD Biosciences). Data analyses were performed using FlowJoTM software (BD Biosciences), and results were presented as graphics where the FMO is overlaid with the stained sample for each antibody/marker used.

### 2.3. hBM-MSC Lentiviral Transduction and Bioluminescence Signal Analysis

The lentiviral vector pMSCV Luc2 T2A Puro (kindly provided by Dr. Deivid de Carvalho Rodrigues) was used to genetically modify the hBM-MSCs to express luciferase using a vesicular stomatitis virus G glycoprotein (VSV-G). The bioluminescent reporter gene for luciferase-2 was encoded by this vector, which was derived from pseudotyped viruses, as well as the puromycin gene N-acetyl-transferase, which provides puromycin resistance. The production of virions used in lentiviral transduction was performed as previously described [13,43,44]. In brief, 2 × 10^5^ hBM-MSC per well (24-well plate) were maintained in DMEN/F-12 medium with 10% FBS until it reached 70% confluence. Then, for transfection, an overnight incubation was performed using a fresh culture medium containing lentiviral vector at 10 multiplicities of infection (MOI) with 8 µg/mL of polybrene (Sigma-Aldrich, Saint Louis, MO, USA). Next, after the overnight incubation, to eliminate viral particles, the medium was exchanged. The transduced hBM-MSC selection initiates 48 h later, with the addition of 1 μg/mL puromycin every two days for one week. As a result, only hBM-MSC expressing luciferase protein and puromycin-resistant (hBM-MSC) remained in the culture [41].

The hBM-MSC Luc bioluminescence signal analysis was performed in 03 different cell concentrations: 10^4^, 5 × 10^5^, and 10^6^, placed in triplicates into a black 96-well plate and using IVIS^®^ Lumina LT Series III equipment (Xenogen Corp, Alameda, CA, USA). The intensity of BLI was detected using the following parameters before and after the addition of 20 μL of D-luciferin (150 mg/mL) (XenoLight, Perkin Elmer, Boston, MA, USA): automatic exposure time, F/stop 4, binning of 4 and FOV of 5 mm with 6 min interval between each image acquisition, over 480 min. The kinetics of the BLI were registered and analyzed in absolutes units (photons/s) using Living Image software version 4.7.3 (IVIS Imaging System, Boston, MA, USA).

### 2.4. Multifunctional Nanoparticles with Magnetic and Fluorescent Properties

The hBM-MSCs were labeled with multifunctional nanoparticles (MFNP) with magnetic and fluorescent properties, which are composed of crystalline iron oxide (Fe_3_O_4_) nucleus coated with dextran. These MFNP have an average hydrodynamic size of 35 nm and a zeta potential of +31 mV. The MFNP were conjugated with two fluorophores whose excitation/emission wavelengths are 750/777 nm (NIR spectrum) and 558/580 nm (visible spectrum) (Biopal, Molday ION™, Worcester, MA, USA).

### 2.5. Multifunctional Nanoparticle Characterization for Optical Properties, Hydrodynamic Size, and Colloidal Stability

The MFNP characterization was performed using cell culture under similar conditions, at a concentration of 10 µg Fe/mL dispersed in DMEN/F-12 medium, supplemented with 10% FBS, and 1% antibiotic-antimycotic solution.

The MFNP optical characteristics were measured using an RF-6000 spectrofluorophotometer (Shimadzu, Kyoto, Japan) and LabSolutions RF software (Shimadzu, Kyoto, Japan). MFNP were dispersed in supplemented medium enclosed in a quartz cuvette with 1 cm of the optical path. The 2D excitation spectrum graphics were obtained in the wavelength between 400 to 850 nm and emissions from 450 to 870 nm.

The hydrodynamic size of MFNP was determined using the dynamic light scattering (DLS) technique with the Zetasizer Ultra system (Malvern, Worcestershire, UK). The polydispersion curve of hydrodynamic size was obtained at an angle of 173°, with the number of averages set at 30 and acquisition time of 3 s in a fixed position at 37 °C, with 120 s of thermic equilibrium. The mean diameter and standard deviation were calculated by adjusting the experimental data to a log-normal distribution function.

The colloidal stability analysis of hydrodynamic diameter over time was performed with MFNP dispersed in the supplemented medium. The polydispersion curves were recorded every hour for 18 h at 37 °C.

### 2.6. Strategies to the Cell Labeling with MFNP Regarding their Concentration, Period of Incubation and Magnetic Field

The cell labeling process was carried out using three MFNP concentrations (10, 30, and 50 µg Fe/mL) dispersed in medium, submitted or not to a static magnet field (190 mT) for 3, 6, 8, 12, 18, and 24 h to determine the best labeling strategy in terms of MFNP concentration, use of magnetic field and period of incubation.

For different labeling procedures, triplicate samples for each condition were gathered. A spherical glass coverslip with a diameter close to the good diameter (10 mm) was placed at the bottom of each well before adding the cells, and this coverslip was used for the MFNP internalization analyses. The 24-well plate was used to culture MSCs until they reach 70% confluent.

Next, the plate was rinsed with PBS, and fresh media with all pre-determined MFNP concentrations was applied for each inoculation time specified above. A neodymium’s magnet (190 mT) was positioned or not positioned underneath the plate providing a static magnetic field.

After the incubation period, the plate was washed three times with PBS to remove all non-internalized MFNP.

### 2.7. Evaluation of MFNP Internalization into Cell by Optical and Fluorescence Microscopy

The internalization of MFNP into cells was examined using bright-field and fluorescence microscopy, as detailed in a previous study [41], according to magnetic (iron oxide) and fluorescent (Rhodamine-B) characteristics of MFNP.

Briefly, the coverslips containing hBM-MSCs labeled with MFNP by all strategies described above and their controls were fixed with 4% paraformaldehyde. Next, they were Prussian blue-stained to highlight the iron present in the cells that internalized nanoparticles. Thus, cells were stained using a solution of 0.25 mg potassium ferrocyanide (K_4_Fe(CN)_6_) (Sigma–Aldrich, Saint Louis, MO, USA) and 5% hydrochloric acid (Merck, Darmstadt, Germany) for 5 min, following by PBS washing.

For fluorescence evaluation, a 4,6-diamidino-2-phenylindole dihydrochloride (DAPI) solution was added for 5 min to cells, followed by PBS washing.

The images were then obtained using Nikon TiE microscopy (Nikon, Tokyo, Japan) with an excitation/emission filter of 358/461 nm for DAPI and 530/550 nm for Rhodamine-B present in the MFNP.

### 2.8. Cellular Viability Determined by BLI after hBM-MSC Internalization of MFNP

The cell viability was evaluated in all labeling process conditions described in Section 2.6 by BLI. Previously, hBM-MSC labeled with MFNP were trypsinized and transferred to a 96-well black plate. The BLI images were acquired after the addition of 20 µL of d-luciferin in each well (150 mg/mL) (XenoLight, Perkin Elmer, Boston, MA, USA) through the IVIS^®^ Lumina LT Series III equipment with the following parameters: automatic exposure time, F/stop 4, binning 4, and a field of view (FOV) of 5 mm. The BLI intensity (photons/s) was selected from a region of interest (ROI) of 2 cm^2^. The cellular viability determination was obtained by the following formula (BLI intensity of hBM-MSC labeled/BLI intensity of hBM-MSC unlabeled) × 100.

### 2.9. Quantification of MFNP Internalized into hBM-MSC Using NIRF and ICP-MS

The MFNP internalized into hBM-MSC was also quantified by the intensity signal of NIRF and Inductively Coupled Plasma Mass Spectrometry (ICP-MS), in all established conditions, at three MFNP concentrations (10, 30, and 50 µg Fe/mL) dispersed in the medium, submitted or not to a static magnetic field (190 mT) for 3, 6, 8, 12, 18, and 24 h of incubation time.

#### 2.9.1. Quantification of MFNP Internalized into hBM-MSC by NIRF Imaging

The same cell samples that were transferred to the 96-well black plate for BLI analysis were used for NIRF internalization evaluation with the IVIS equipment through the same parameters described in item 2.8, with an excitation filter of 710 nm and emission of 780 nm. The NIRF intensity (photons/s) was selected from a region of interest (ROI) of 2 cm^2^.

#### 2.9.2. Quantification of MFNP Internalized into hBM-MSCLuc by ICP-MS

The mass spectroscopy technique was used to determine the quantity of iron incorporated into hBM-MSC by MFNP labeling utilizing the Inductively Coupled Plasma Mass Spectrometry (ICP-MS) model Nexion 350× (Perkin Elmer, Boston, MA, USA).

For this analysis, samples containing 1 × 10^6^ cells/mL labeled with MFNP by all internalization strategies (as described in item 2.6) were dispersed in 100 µL of PBS and 300 µL of nitric acid (37%) for digestion over 4 h at 70 °C.

To evaluate the iron content in the samples, they were diluted 200 times with Milli-Q^®^ water and analyzed by ICP-MS. The measurements were performed in triplicate, and the quantification was completed using a calibration curve and certified standard iron (NexION # N8145054).

### 2.10. Focal Ischemic Stroke Model, Blood Perfusion Evaluation and Experimental Groups

Male Wistar rats aged 2 months (weighed from 250 to 300 g) were kept at the Hospital Israelita Albert Einstein—Animal Experimental Surgery and Training Center (CETEC) vivarium and were exposed to 21 ± 2 °C room temperature, 60 ± 5% relative humidity in full ventilation, 12 h light/dark cycle (7 a.m.–7 p.m.), and ad libitum access to food and water. The vivarium was accredited by the Association for Assessment and Accreditation of Laboratory Animal Care International (AAALAC International), and the general conditions were assessed daily. The study was approved (number 3069-18) by the Committee in Ethics for Animal Research of the Hospital Israelita Albert Einstein (São Paulo, Brazil).

The focal ischemic stroke was induced using the photothrombotic model as previously described [41]. Rose Bengal (50 mg/mL) (Sigma-Aldrich) was injected intraperitoneally into the bloodstream followed by laser application (light source, Hamamatsu) positioned 4–5 mm from the dura mater, under the pre-central gyrus for 65 min with 50% laser power.

The focal ischemic brain lesion was confirmed by local blood perfusion analysis with a PeriCam Perfusion Speckle Imager (PSI) system (Perimed, Stockholm, Sweden), comparing the baseline of blood perfusion to ischemic areas perfusion of left somatosensory cortex after ischemia induction (around 70% of blood decrease was considered effective ischemia), this condition and procedure were described in previous studies [41,45].

The experimental design included three groups, and all animals were submitted to stroke induction and evaluated after perfusion during a surgical procedure. After 24 h of surgery, the control group received unlabeled hBM-MSC, group A received hBM-MSC labeled with MFNP without a magnetic field, and group B received hBM-MSC labeled with MFNP in the presence of the static magnetic field. The period of incubation used was the best one for cell labeling analyzed in Section 2.6.

The animals were anesthetized with the same procedure described above, the surgical incision was reopened, and the cells were slowly administered around the ischemic brain lesion area using a Hamilton syringe (total of 30 µL administered for 10 min); the procedure was finalized with the suture of surgical incision.

The animals were placed on a heating pad throughout anesthesia to keep their rectal temperature at 37.0 °C at the end of all surgical procedures (PhysioSuite, Kent Scientific Corporation, Torrington, CT, USA).

### 2.11. hBM-MSC Labeled with MFNP Homing Evaluation after In Vivo Implantation by NIRF and BLI

Cells homing were assessed in vivo using BLI and NIRF images at 1 and 7 h after implantation, with the surgical incision open to reduce signal attenuation. All animals received D-luciferin (150 mg/kg) intraperitoneally 10 min before BLI acquisition.

The rats were then euthanized, and the entire brain was extracted for ex vivo image capture (around 7 h after cell implantation). The BLI and NIRF images were obtained with the same parameters described in Section 2.8 and Section 2.9.2 by IVIS equipment. The intensity signals of both techniques were quantified by in vivo images according to the best condition for MFNP concentration and incubation determined by the in vitro analysis described previously in Section 2.8 and Section 2.9.

### 2.12. Statistical Analysis

Data were displayed as mean and standard deviation and Levene’s test was used to determine variance equality. BLI and NIRF intensity signal quantification was compared using a two-way ANOVA test, followed by a post-hoc comparison corrected by Bonferroni. All analysis was considered at the 0.05 level of significance. JASP software v0.16 (http://www.jasp-stats.org, accessed on 18 February 2022) was used for all statistical analysis.

## 3. Results

### 3.1. hBM-MSC Characterizations

The hBM-MSCs unlabeled and labeled with 50 µg Fe/mL of MFNP used in this study were uniform in size and granularity (FSC vs. SSC). They did not express markers for endothelial cells (CD31 and CD106). These cells also did not express hematopoietic markers such as CD14, CD19, CD34, CD45, or HLA-DR, they expressed CD29, CD44, CD73, CD90, and CD105 (Figure 1). They displayed a fibroblastic shape and were plastic adherent. Thus, we concluded that, as expected, hBM-MSCs displayed stromal cell characteristics that fulfill the international phenotypic criteria determined by the International Society for Cellular Therapy and the labeling with 50 µg Fe/mL of MFNP was unable to alter their phenotype or primary characteristics [46].

### 3.2. MFNP Characterization and Internalization Analysis into hBM-MSC by Fluorescence and Brightfield Microscopy

MFNP fluorescence and optical properties were evidenced by two peaks of intensity in the excitation/emission wavelength of 557/604 nm for Rhodamine-B (fluorophore that corresponds to the region of the visible spectrum) and 757/780 nm for fluorescent-750 nm (fluorophore corresponding to the region of the infrared spectrum) (Figure 2I), according to the presence of both fluorophores coupled to the MFNP surface.

The values found matched the infrared spectrum and implied good applicability for the in vivo experiments; due to the low absorption of biological molecules in this part of the spectrum, and it is also applicable for in vitro experiments using the visible spectrum.

The MFNP mean hydrodynamic diameter (dispersed in culture medium) was 58 nm (Figure 2II) according to the polydispersion size analyzed for MFNP after adjusted to log-normal distribution. The value of 58 nm is higher than the values indicated in the Technical Information of this nanoparticle (35 nm). Furthermore, the stability of hydrodynamic size (polydispersion) over time showed to be adequate, without the presence of agglomerates during 18 h of analysis (Figure 2III)

Figure 2IV displays the evaluation of the MFNP internalization process into cells, using the concentration of 10 µg Fe/mL with or without magnetic field during different periods of incubation (3, 6, 8, 12, 18, 24 h) by fluorescent and brightfield microscopy images (amplified in 10× and 40×). The fluorescence images highlighted the red color, confirming the presence of Rhodamine-B fluorophore coupled with MFNP internalized into hBM-MSC cytoplasm and the blue color, which showed the cell’s nucleus stained by DAPI. The bright-field microscopy images confirmed the fluorescence analysis since they were obtained from the same area and display the MFNP internalized into hBM-MSC cytoplasm by MFNP stained in Prussian blue.

Visually, during all periods of incubation analyzed, it was possible to observe a signal in the labeled cells without the presence of a magnetic field (Figure 2IV(A–L)) which was a lower signal compared to cells labeled in the presence of a magnetic field (Figure 2IV(M–X)), these findings were confirmed by the presence of fluorescence and Prussian blue images.

Periods of incubation of 3 to 12 h resulted in a significant increase in signal intensity (Figure 2IV(A–H,M–T)) compared to long periods of incubation. Further, periods over 18 and 24 h resulted in a less evident increased in signal intensity (Figure 2IV(I–L,U–X)). Thus, the best increase in signal intensity was noted in early periods of incubation ranging from 3 to 12 h.

Images (fluorescence and brightfield) comparing two concentrations of MFNP (30 and 50 µg Fe/mL) internalization into hBM-MSC are displayed in Figure 3I,II. The signal intensity was less evident in cells labeled without the presence of a magnetic field (Figure 3I(A–L)) compared to cells labeled in the presence of a magnetic field (Figure 3II(M–X)). This result confirms the importance of a magnetic field for the best results in cell labeling and obtaining a higher signal intensity.

hBM-MSCs labeled with 30 µg Fe/mL of MFNP showed increased signal intensity at incubation times up to 8 h (Figure 3I(A–F,M–R)), and for cells labeled with 50 µg Fe/mL of MFNP, this increase was visible up to 6 h (Figure 3II(A–D,M–P)); after these times, the signal intensity in both fluorescence and brightfield images remained uniform. This result demonstrated that in using a higher concentration of MFNP as 50 µg Fe/mL, the best results are obtained in a shorter period of time.

### 3.3. Cellular Viability after MFNP Internalization Process into hBM-MSC by Different Strategies

The bioluminescence kinetic evaluation of cell transfection by cell concentration was performed after the lentiviral transduction procedure into hBM-MSC. As shown in Figure 4A, this evaluation showed the maximum signal BLI (21.02 ± 1.0) × 10^8^, (10.12 ± 0.5) × 10^8^, and (0.35 ± 0.1) × 10^8^ photons/s corresponded to 1 × 10^6^, 5 × 10^5^, and 1 × 10^4^ hBM-MSC concentrations. This result demonstrated a direct correlation between the cell’s concentration and signal intensity.

The bioluminescence technique was used to evaluate the cellular viability based on the signal intensity of transfected hBM-MSCs after MFNP labeling at all different conditions proposed, as shown in Figure 4B. The analyzed results from Figure 4B are displayed in Figure 4C and Table 1.

Thus, in the absence of a magnetic field, cellular viability decreased 2.0% for cells labeled with maximum MFNP concentration (50 μg Fe/mL) and for longer periods of incubation (24 h), whereas in the presence of a magnetic field, viability declined in 4.0%. Comparing both magnetic field conditions up to 8 h of incubation, the maximum decrease of cellular viability was less than 2.1% for all MFNP concentrations.

Cells labeled with 50 μg Fe/mL of MFNP in the presence of magnetic field within 18–h and 24 h of incubation period were the ones that displayed the highest decrease in viability (3.6–4.0%).

### 3.4. Quantification of MFNP Internalized into hBM-MSC by NIRF and ICP Techniques

The quantification of MFNP internalized into hBM-MSC was performed indirectly using NIRF, and directly using ICP-MS (Figure 5).

The indirect quantification of MFNP was performed by measuring the iron load by the NIRF intensity in all conditions, as displayed in Figure 5A, the signal intensity was correlated with different grades of red-yellow colors, ranging from the lower intensity in dark red to the medium in red until the highest in yellow.

Figure 5B depicts the NIRF signal intensities associated with exponential adjustments for each condition, demonstrating that the high intensity of iron load is directly proportional to the increased periods of incubation and the highest MFNP concentration.

However, the magnetic field also had a high impact on the iron load. The NIRF signal intensity of the hBM-MSCs labeled with 10 μg Fe/mL at 24 h without a magnetic field application was 2.00 ± 0.10 × 10^8^ photons/s was close to the NIRF intensity of the cells labeled with the same MFNP concentration with 3 h of incubation, in the presence of a magnetic field (2.21 ± 0.08× 10^8^ photons/s). Once comparing the use of a magnetic field in the same period of incubation (24 h), NIRF signal intensity was about four times higher (8.09 ± 0.39 ×10^8^ photons/s) in the presence of a magnetic field, as shown in Table 2.

Thus, the NIRF signal intensity analysis based in the period of incubation revealed a progressive increase according to timing until 12 h and MFNP concentrations. Yet, the presence of a magnetic field during the labeling process increased the NIRF intensity up to 3.3 to 5.6 times compared to the absence of a magnetic field, as shown in Table 2 and the continuous line curves versus traced line curves in Figure 5B, as also in the boxplot in Figure 5E.

For direct quantification, a calibration curve (Figure 5C) was used to determine the MFNP iron load in terms of mass (*p*g Fe/cells) and the number of nanoparticles per cell ×10^5^, as shown in Figure 5D and Table 3.

The experimental data were adjusted by performing an exponential modification represented by the curves shown in Figure 5D for each labeling condition tested (by concentrations of MFNP, presence or absence of a magnetic field, and over the period of incubation). When comparing indirect quantification (NIRF—Figure 5B) and direct quantification (ICP-MS—Figure 5D), the curves that indicate the internalization of MFNP into hBM-MSC displayed a similar trend pattern.

The direct iron load quantification after the cellular labeling process without the presence of a magnetic field showed a range from 0.64 to 8.00 *p*g Fe/cell (0.81 × 10^4^ to 10.05 × 10^4^ MFNP per cell), according to MFNP concentrations and incubation period, as shown in Table 3. When this evaluation was performed after cell labeling in the presence of a magnetic field, there was a remarkable increase in iron internalized, ranging from 2.21 to 29.17 *p*g Fe/cell (2.78 × 10^4^ to 36.65 × 10^4^ MFNP per cell), corresponding to 13 times higher the iron load compared to the previous condition (Table 3). In the NIRF quantification, the signal intensity range was from 2.21 to 25.23 photons/s, which corresponded to a signal increase of 11.41 times (Table 2).

Yet, there was an increase in iron load using magnetic field during labeling according to different concentrations of MFNP, as shown in the boxplots of Figure 5E and Table 3.

The distribution of the iron load in the presence or absence of a magnetic field of each labeling condition was analyzed by both indirect measurement (NIRF, Figure 5B) and direct measurement (ICP-MS, Figure 5D) as shown in the box plots in Figure 5E. The data dispersion was higher in the indirect measure (NIRF), showing high amplitude for data distribution compared to direct measures (ICP-MS).

The influence of iron load internalized into hBM-MSC as a function of incubation time $C_{MFNP}\left(t\right)$ can be analyzed through the following relationship formula:(1)$$C_{MFNP}\left(t\right)=C_{MFNP}^{max}\left[1-e^{-\frac{t}{\tau_{MFNP}}}\right]$$
where $C_{MFNP}^{max}$ is the maximum concentrations of nanoparticles internalized by MSCs under the given conditions. The $\tau_{MFNP}$ is the characteristic time of MFNP internalization into MSC that corresponds to the time iT requires to internalize 63% of the maximal concentration.

After the curves adjustment of Figure 5D following the relation (1), the obtained data were displayed in Table 4.

The maximum MFNP concentration that can be achieved in hBM-MSC internalization process when labeled without a magnetic field with 10, 30, and 50 µg Fe/mL was 2.50 ± 0.15, 7.14 ± 0.32, and 7.56 ± 0.68 *p*g Fe/cell respectively. For the same process in the presence of a magnetic field, the maximum concentration was 11.10 ± 0.19, 31.43 ± 2.30, and 35.24 ± 0.15 *p*g Fe/cell. This result demonstrated an increase of 4.5 times for all concentrations when comparing the presence of the magnetic field with their absence. The $\tau_{MFNP}$ ranged from 4.28 to 8.83 h to achieve 63% of the maximum MFNP concentration.

The highest iron load by hBM-MSC was reached at a concentration of 50 µg Fe/mL in the presence of the magnetic field. According to the adjustment of the curves in Figure 5D, it was verified that the incubation time required to reach 90% of the maximum concentration of MFNP was approximately 10.5 h, with an estimated iron load of 26.60 *p*g Fe/cell.

When cellular toxicity was investigated (Figure 4), it was possible to observe that after 9 h of incubation, there was a dramatic reduction in cell viability compared to the other labeling conditions, reaching a toxicity of about 4%. Therefore, the recommended incubation time was 9 h, which corresponded to a 25.4 *p*g Fe/cell iron load (86% of the iron load internalized in 24 h).

For the in vivo study, were used three conditions, one as a control without iron load and two others with 9 h of incubation and 50 µg Fe/mL of MFNP concentration, varying only for the presence or absence of a magnetic field.

### 3.5. In Vivo Evaluation of hBM-MSC Labeling with the Best Condition of MFNP Internalization by Intensity Signal of BLI and NIRF, after Confirming the Focal Ischemic Brain Lesion

The focal ischemic brain lesion was confirmed by the decrease in 67% of blood perfusion, comparing the basal perfusion (before ischemia induction) to the same region, after stroke induction, as shown in the graphic of Figure 6I(A).

The perfusion data were measured using the same region of interest (ROI), delineated manually and used in both images, before (Figure 6I(B)) and after (Figure 6I(C)) ischemic induction. The left medial cerebral artery crosses the ROI and their aspect before and after ischemia induction can be seen in Figure 6I(D,E), and the laser position before calibration and during ischemic induction was shown in Figure 6I(F,G).

hBM-MSCs were implanted in vivo after 24 h of ischemia brain induction, using hBM-MSCs prepared in 03 different conditions, one as the control without iron load, and 2 others with 9 h of incubation and 50 µg Fe/mL of MFNP concentration, varying only for the presence or absence of a magnetic field. The implantation was followed by evaluation using BLI (Figure 6II) and NIRF (Figure 6III) intensity signals at 1 and 7 h after implantation, aiming to evaluate hBM-MSC homing at acute and subacute stages.

Since the hBM-MSCs were genetically modified to express luciferase, the BLI signal was observable in all three conditions (Figure 6II(A–I)), and the signal was retained in the local brain injury until 7 h after cells implantation (Figure 6II(D–F)), with enhanced visualization in the ex vivo BLI image (Figure 6II(G–I)).

The quantification of BLI intensity signal revealed a similar pattern for all 03 conditions tested at 1h, as well as at 7 h after implantation, but with a discreet increase in signal strength over time, as shown in the histogram bars in Figure 6IV(J).

The labeling did not affect the BLI signal. Furthermore, the increase in the BLI signal over time also demonstrated that cell handling had no effect on their survivability or growth during 7 h, which was the period of evaluation.

The NIRF signal intensity was detectable only in hBM-MSCs labeled with MFNP (Figure 6III, except A, D, and G), as expected, due to the presence of a fluorophore coupled on the MFNP surface.

The images of NIRF intensity were visually higher in the animal’s that received hBM-MSCs labeled in the presence of a magnetic field (Figure 6III(C,F,I)) compared to animals that received hBM-MSCs labeled in the absence of a magnetic field (Figure 6III(B,E,H)), the signal visualization was more intense in the ex vivo images (Figure 6III(G–I)).

NIRF signal intensity quantification corroborated the differences described above (*p* < 0.001). For hBM-MSCs labeled with the magnetic field, the NIRF signal at 1 and 7 h was 1.8 × 10^9^ and 1.6 × 10^9,^ respectively; for hBM-MSCs labeled without a magnetic field, the NIRF signal at 1 and 7 h was 5.7 × 10^8^ and 5.3 × 10^8^, respectively, as demonstrated in Figure 6IV(K).

Furthermore, there was only a discreet decay in the NIRF signal intensity for the two evaluated conditions over time.

## 4. Discussion

One of the most important factors to be considered when evaluating MSC therapy in neurological diseases is the delivery of cells to the brain injury site [47]. In this regard, there are several in vitro and in vivo approaches that can be used to evaluate the therapeutic process, including aspects of cell homing and tracking [21,28,48,49].

These approaches can include the use of contrast agents [13,20,42,50] associated with molecular imaging techniques, which can increase the sensitivity of signals captured through different strategies [51,52].

In this context, this study demonstrated that for spectroscopic evaluation, the multimodal nanoparticles (magnetic and fluorescent) used in cell labeling were more efficiently internalized using static magnetic fields with MFNP concentrations and period of incubation optimized. These results corroborated NIRF signal detection and were confirmed by an in vivo study in a stroke model, which demonstrated good signal intensity without compromising cell viability.

The characterization of MFNP dispersed in culture media confirmed a good MFNP colloidal stability over 18 h, retaining the hydrodynamic diameter of 58 nm, an important aspect of the cell labeling process [13,53,54]. The imbalance of forces involved in the interaction of nanoparticles with the medium can lead them to agglomerate [35,55] and thus to their non-internalization and toxicity for hBM-MSC.

The study of Wiogo [54] demonstrated the importance of FBS percentage in the culture medium for colloidal stability, particularly for the maintenance of ionic and magnetic forces, and that the FBS concentration close to 10%, the one used in this study, was the most suitable to maintain stability, providing protein adsorption on their surface. In addition, for clinical phases or more advanced phases of product development, the FBS can be replaced for human pooled AB serum, improving the safety of this therapy or other therapies using MFNP.

Another important physical-chemical feature in the internalization of MFNP is their positive zeta potential of 32 mV; when interacting with the hBM-MSC, which has a negative membrane charge, the positive MFNP feature promotes enhanced nanoparticle internalization due to the electrostatic process involved [13,55,56,57].

The optical properties of magnetic nanoparticles are due to the presence of two fluorophores coupled on their surface, allowing in vitro (visible length fluorescence) and in vivo (NIRF) measurement of the labeling process.

Fluorescence microscopy revealed differences in cell labeling when performed with different concentrations of MFNP (10, 30, and 50 µg Fe/mL) or varied periods of incubation (from 3 to 24 h), as well as the presence or absence of a magnetic field (190 mT).

There was a positive correlation between increased fluorescence intensity and period of incubation, which was intensified by the application of a static magnetic field, a phenomenon also directly related to the MFNP concentration.

According to Chen et al. [58] and Yun et al. [52], the period of incubation ranging from 3 to 12 h resulted in a considerable increase in signal intensity. However, periods of incubation from 18 to 24 h did not clearly improve the previous results.

Using higher MFNP concentration, the increase in signal intensity was evident in short periods of incubation, as 30 µg Fe/mL of MFNP provided better signal intensity at 8 h, while for 50 µg Fe/mL, the best signal was achieved at 6 h. These findings were supported by bright field images captured from the same area as the fluorescence analysis, which highlighted the blue color of the iron oxide nanoparticles.

Fluorescence and brightfield microscopy are well-established techniques for evaluating the process of nanoparticle internalization in cells, and cumulative periods of incubation increase signal intensity, as shown in studies where fluorescent and magnetic properties of nanoparticles were used to evaluate in vitro studies [34,52,56,58,59,60].

The cell labeling strategies used in this study did not significantly interfere with cell viability, even when using the highest concentration of MFNP (50 μg Fe/mL), and the longest period of incubation (24 h). The presence of a magnetic field reduced the viability decreased from 100% to 96%, and in the other conditions, the reduction was to 98%, thus, this type of MFNP showed low toxicity when compared to MFNP used in other studies [52,61,62,63].

A study using 20 µg Fe/mL NP reported higher cellular toxicity of almost 50% [52]. This high toxicity can be related to several factors such as NP coating, its degradation products, size and geometry, the cellular environment chemical composition, the oxidation state of iron in SPION, and protein-SPION interactions [62,63,64].

In addition to the qualitative analysis of the labeling process, an indirect quantitative analysis was also performed of the labeled cells’ using NIRF signal for different labeling strategies in in vitro, as well as the direct method of ICP-MS quantification of the iron load internalized by cells.

This study also highlighted that the magnetic field used during the labeling process enhanced the NIRF signal strength from 3.3 to 5.6 times compared to labeling strategies that did not use a magnetic field, besides maintaining the signal intensity in shorter periods of incubation.

The iron load quantification, for MFNP highest concentration, by ICP-MS was 8.00 *p*g Fe/cell (10.05 × 10^4^ MFNP per cell) without a magnetic field, and 29.17 *p*g Fe/cell (36.65 × 10^4^ MFNP per cell) with a magnetic field. This result without a magnetic field was very close to our group’s previous results of 6.71 *p*g Fe/cell (8.43 × 10^4^ NP per cell), which used the same cell labeling process for 18 h [13].

Other studies adopting even higher MFNP concentrations (100 μg Fe/mL) for the cell labeling process and direct quantification by ICP-MS achieved 6.9 *p*g Fe/cell [65] and 5.3 *p*g Fe/cell [58] using neuronal progenitor cells and 48 h of incubation achieved similar results. Using the same NP concentration in the cell labeling over 24 h, MRI quantification of iron load yields near values of 5 *p*g Fe/cell (6 × 10^4^ NP per cell) [66].

The study by Landazuri [34] found that MSCs labeled with MFNP in the presence of magnetic field displayed an iron load of ~9.4 *p*g/cell (1.48 × 10^6^ SPION/cell). These results support our findings that the magnetic field increases the iron load by cells. Further, the effectiveness of the cell labeling process with the magnetic field, also called magnetofection, has been demonstrated in other studies employing several cell types [67,68,69,70,71,72,73].

Another essential part of the cell labeling process assessment was the period of incubation; the iron load internalized into hBM-MSC was time-dependent, and the magnetic field presence led to outstanding outcomes for MFNP internalization.

The most frequent period of incubation for cell labeling reported in the literature [41,53,66,74,75,76,77,78,79,80] was 18 h. Herein, for the same period of incubation, an iron load of 36.79 *p*g Fe/cell was obtained using the highest nanoparticle concentration (50 µg Fe/mL) in the presence of a magnetic field.

In the study by Yun [52], using 15 µg Fe/mL nanoparticle for cell labeling process and incubation periods of 3, 6, and 24 h, the Fe load in 6 h was 71.4% of the Fe load in 24 h, with cell viability of around 90%.

Regarding the labeling process over cell viability in this study, it was determined an optimal incubation period of 9 h, yielding an estimated iron load of 25.4 *p*g Fe/cell, which was higher than reported previously in the literature [13,65,77,80,81,82,83].

In summary, this study demonstrated higher MFNP internalized in hBM-MSCs with low toxicity while preserving their immunophenotypic properties in a shorter period of incubation.

To validate our results, the hBM-MSC were administrated in vivo, 24 h after stroke induction, using the best conditions determined in vitro as concentration (50 µg Fe/mL) and 9 h of incubation for the cell labeling process.

After cell implantation, the BLI signal intensity increased over time, showing high cell viability in the brain lesion site, allowing the cell homing analysis through the increased signal intensity during 1 to 7 h (2.8 to 3.0 × 10^9^ photons/s). Additionally, the labeling process did not interfere with cells viability, as only a tiny variation in signal strength was seen (of 2.8 to 2.5 × 10^9^ photons/s), even in the presence of a magnetic field, when the iron load internalization was substantially higher (2.6 × 10^9^ photons/s).

The success of cell homing by BLI has been reported in other studies and displayed the same pattern of increasing BLI signal intensity over time [41,81,82,84]; however, the impact of optimized MFNP internalization by the presence of magnetic field on cell viability in vivo analysis was less investigated.

The NIRF signal intensity was significantly higher on the cells labeled with magnetic field presence, showing a relation of 2.15 times higher at 1 h and 2.0 times at 7 h in comparison to cells labeled without magnetic field. These findings could have improved cell homing and tracking due to the upper spatial resolution and higher sensitivity of the NIRF technique.

Although several researchers have indicated that the use of magnetic fields improves the in vitro labeling process, the evaluation of cell homing and tracking in vivo has received little attention.

The NIRF signal intensity values of hBM-MSC labeled without magnetic field were in the same order (10^8^ photons/s) found in the previous study of our group [41], which used the same type of cells. In other studies, the NIRF technique was also used for cell tracking over time in in vivo study [77,80,85].

Cell homing and tracking evaluation using several modalities of molecular images of the same animal has been possible due to the development of multimodal nanoparticles [30,31,32] and the improvement in the efficient labeling procedures using a magnetic field.

In this study, the magnetic and fluorescent nanoparticles displayed characteristics that allow their evaluation by NIRF and MRI, offering complementary information on physiological and anatomical aspects [23,24], as well as the possibility of evaluation of disease evolution and neurodegeneration after the use of advanced cell therapy.

Regarding the study’s limitations, the signal was evaluated only at two-time points after cell transference in vivo, reducing the other possible conclusions about the cells’ permanence in the lesion or signaling over time. Another limitation of this study was the lack of evaluating of the impact of different concentrations of cells on the detection sensitivity in vivo.

## 5. Conclusions

This study optimized the cellular labeling process in vitro using magnetic field, which resulted in shorter period of incubation with efficient iron load internalization without compromising cellular viability or phenotype. In addition, this optimization process significantly improved cell detection by the NIRF technique. These findings demonstrated the potential of multimodal nanoparticles for future studies to improve diagnosis, therapy, and theranostic evaluation of diseases.

## Figures and Tables

**Figure 1 pharmaceutics-14-01249-f001:**
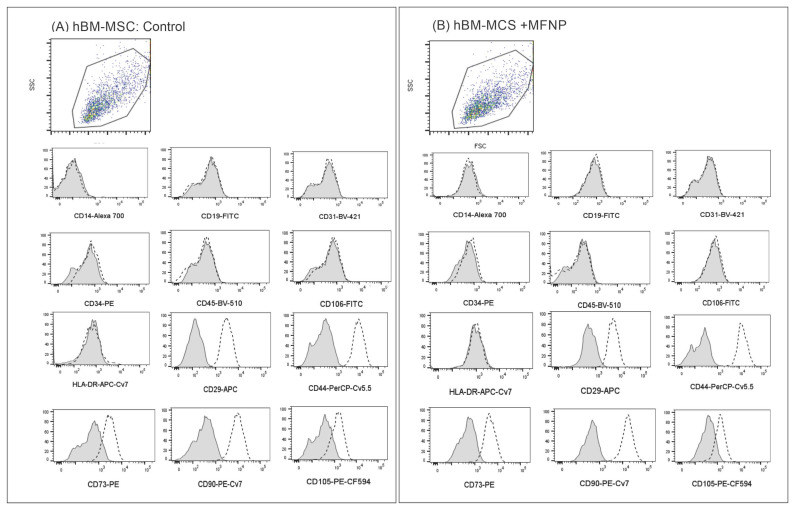
Flow cytometry evaluation comparing the hBM-MSC. (**A**) unlabeled with MFNP and (**B**) labeled with 50 µg Fe/mL of MFNP.

**Figure 2 pharmaceutics-14-01249-f002:**
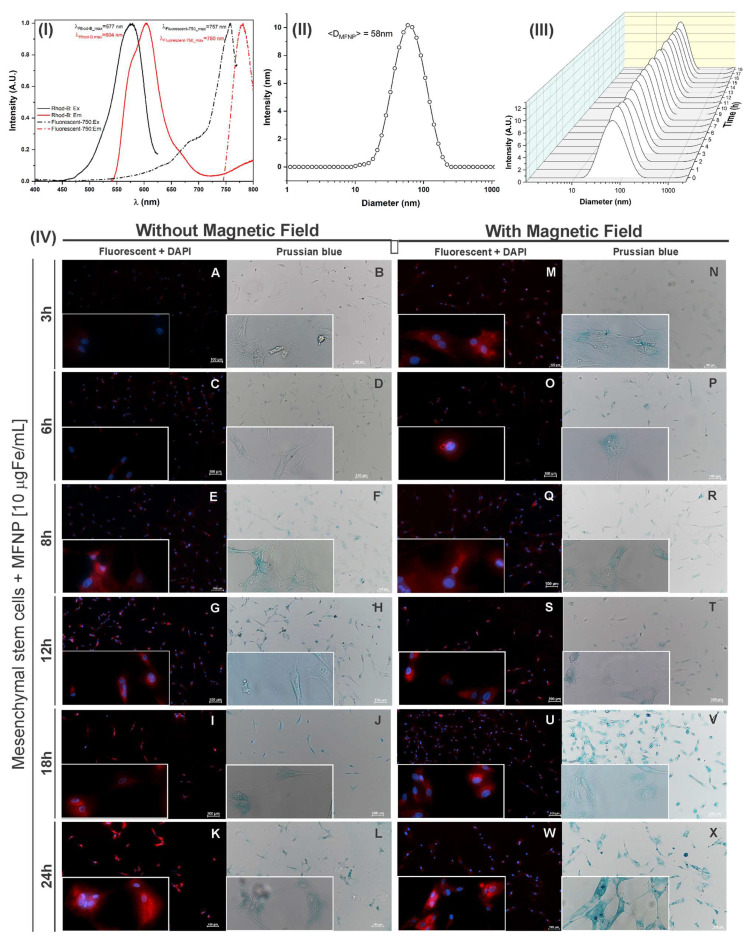
Multifunctional nanoparticle characterization by optical properties (**I**), hydrodynamic size (**II**), and stability (**III**). Microscopy optical images of hBM-MSC labeled with 10 µg Fe/mL of MFNP (**IV**) by fluorescence, highlighting in blue (DAPI), the cell’s nucleus containing the iron of MFNP and the red the MFNP internalized into MSC without magnetic field (**A**,**C**,**E**,**G**,**I**,**K**) and with a magnetic field (**M**,**O**,**Q**,**S**,**U**,**W**), and by brightfield without magnetic field (**B**,**D**,**F**,**H**,**J**,**L**) and with a magnetic field (**N**,**P**,**R**,**T**,**V**,**X**) at different times of incubation. Abbreviations: MFNP—Multifunctional nanoparticle.

**Figure 3 pharmaceutics-14-01249-f003:**
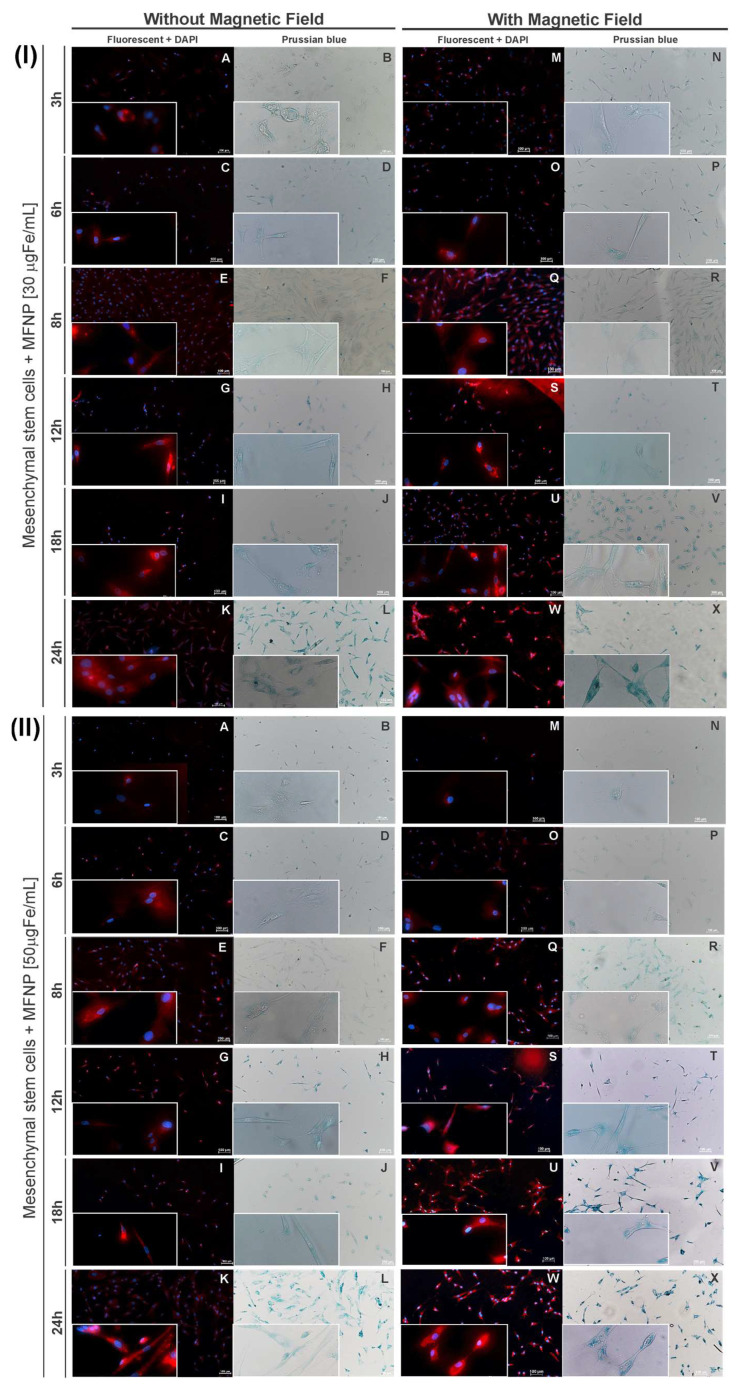
Microscopy images of MSC labeled with 30 (**I**) and 50 (**II**) µg Fe/mL of multifunctional nanoparticles (MFNP) by fluorescence without a magnetic field (**A**,**C**,**E**,**G**,**I**,**K**) and with a magnetic field (**M**,**O**,**Q**,**S**,**U**,**W**). By brightfield without magnetic field (**B**,**D**,**F**,**H**,**J**,**L**) and with a magnetic field (**N**,**P**,**R**,**T**,**V**,**X**). Abbreviations: MFNP—Multifunctional nanoparticle; MSC—mesenchymal stem cell.

**Figure 4 pharmaceutics-14-01249-f004:**
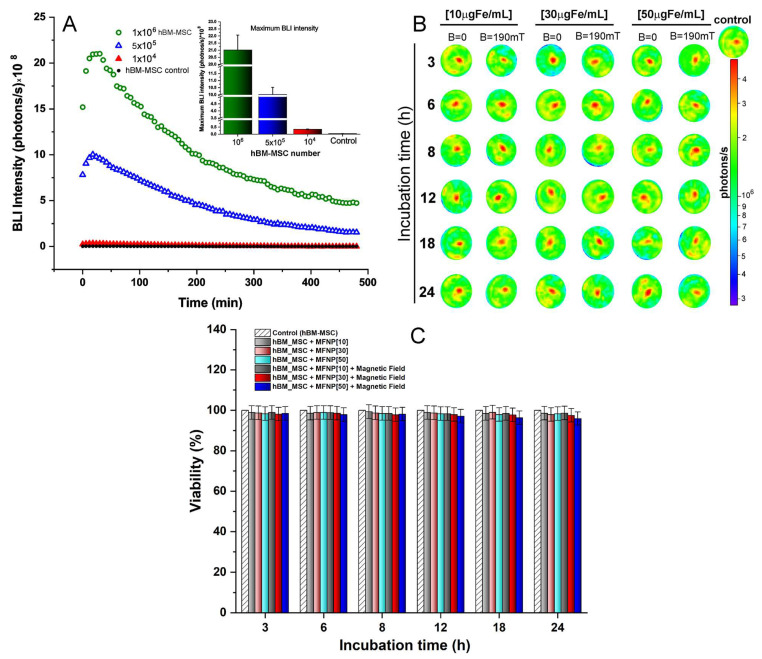
Bioluminescent (BLI) evaluation in vitro. (**A**) The BLI kinetics of the hBM-MSC_Luc_ according to cell concentrations (represented by green, blue, and red symbols) and the respective controls (represented by empty black symbols), following by the BLI signal intensity quantification, shown in the inset graphic. (**B**) The bioluminescence signal intensity of the hBM-MSCs transfected after labeling with MFNP in different concentrations (10, 30, and 50 μg Fe/mL) and periods of incubation (3 to 24 h). (**C**) The cellular viability in all conditions was analyzed.

**Figure 5 pharmaceutics-14-01249-f005:**
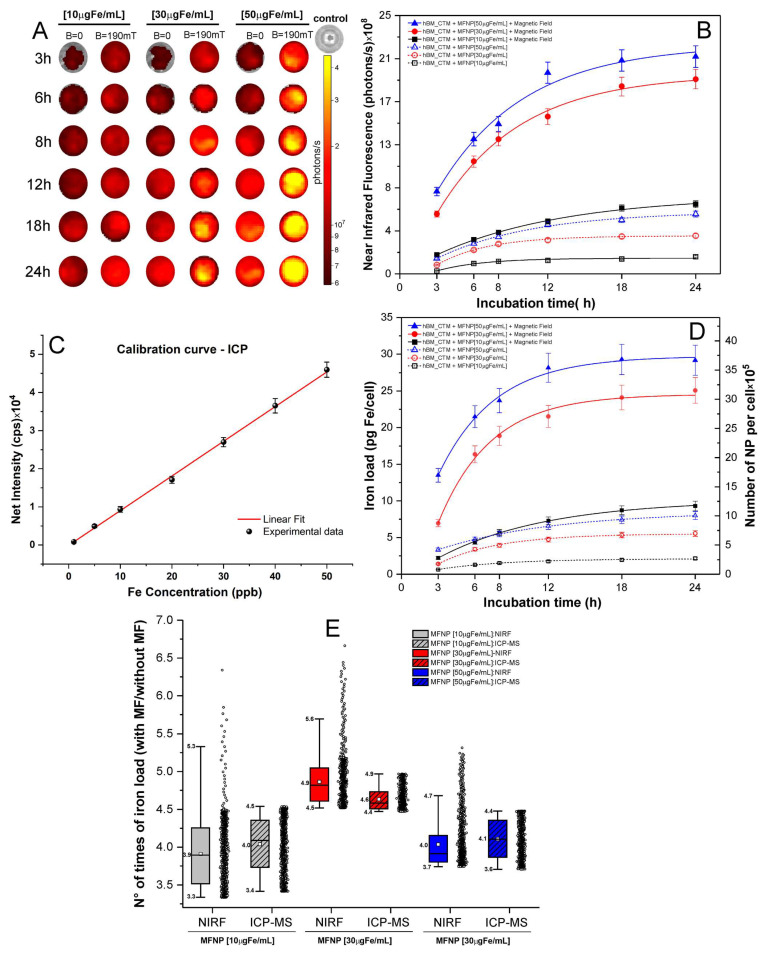
The iron load of MFNP into hBM-MSC quantification by NIRF and ICP, in all labeling conditions tested. (**A**) The NIRF signal intensity analyses over time in different MFNP concentrations and with or without magnetic field (190 mT). (**B**) The iron quantification of NIRF curves in all conditions analyzed, adjusted to lognormal. (**C**) The calibration curve for the ICP quantification; (**D**) The ICP curves of iron quantification of MFNP in all conditions analyzed, and (**E**) the box plot of the NIRF and ICP-MS measure in each condition analyzed with the data represents by the circles.

**Figure 6 pharmaceutics-14-01249-f006:**
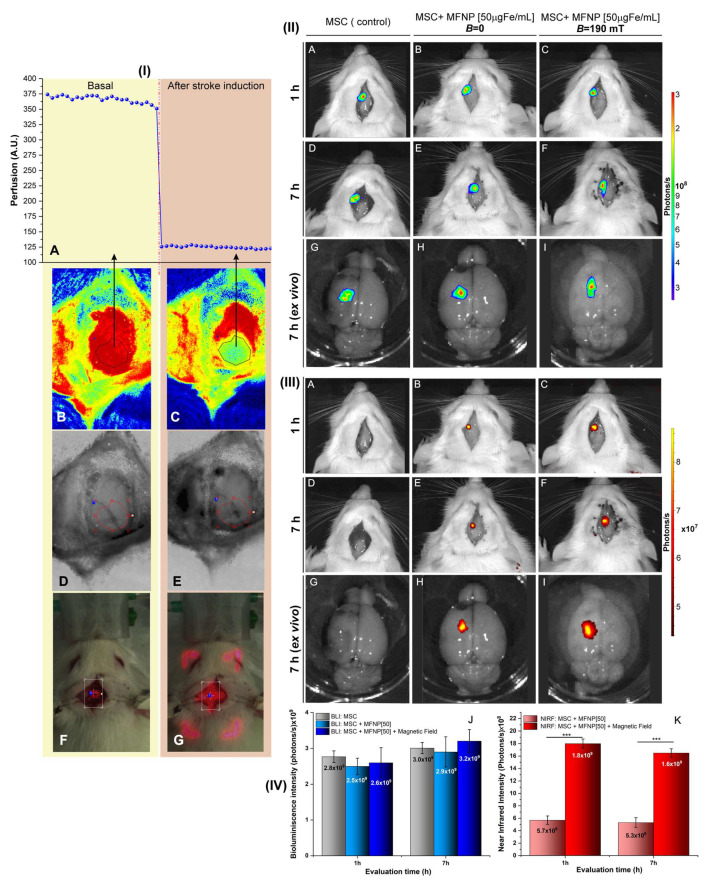
In vivo experiment of the homing of the cell labeled with MFNP by NIRF and BLI images in a focal stroke model. (**I**)—Evaluation of ischemic brain lesion by the decreased blood perfusion (**A**), visually shown in the region of interest (ROI), before (**B**), and after (**C**) photothrombotic induction in the median cerebral artery crossing the ROI (**D**,**E**) through the exposition of a laser to activate the sensitizer dye administrated (**F**,**G**). (**II**)—The comparison of cell homing based on the labeling with the best condition of in vitro experiment with or without submission of a magnetic field by (**II**) BLI and (**III**) NIRF images after 1 h (**A**–**C**), and 7 h of cells implantation (**D**–**F**), then ex vivo images (**G**–**I**). (**IV**)—The quantification of the intensity signal of BLI (**J**) and NIRF (**K**) for all conditions analyses according to the time of image acquisition. *** *p* < 0.001.

**Table 1 pharmaceutics-14-01249-t001:** The percentage of cellular viability after the hBM-MSC labeling with MFNP in the presence or absence of a magnetic field.

Incubation Time (h)	Viability (without Magnetic Field) %			Viability (with Magnetic Field-190 mT) %			Control
	10 mg Fe/mL	30 mg Fe/mL	50 mg Fe/mL	10 mg Fe/mL	30 mg Fe/mL	50 mg Fe/mL	
3	99.0 ± 3.4	98.8 ± 3.4	98.4 ± 3.4	99.0 ± 3.3	98.2 ± 3.3	98.5 ± 3.3	100
6	98.6 ± 3.3	99.0 ± 3.4	99.0 ± 3.3	98.9 ± 3.1	98.6 ± 3.2	98.0 ± 3.4	100
8	99.4 ± 3.4	98.8 ± 3.4	98.5 ± 3.2	98.5 ± 3.4	97.9 ± 3.3	98.2 ± 3.3	100
12	99.0 ± 3.4	98.8 ± 3.4	98.4 ± 3.4	98.4 ± 3.3	98.0 ± 3.4	97.2 ± 3.1	100
18	98.5 ± 3.3	99.1 ± 3.3	98.0 ± 3.4	98.5 ± 3.3	97.8 ± 3.3	96.4 ± 3.2	100
24	98.6 ± 3.3	98.0 ± 3.3	98.4 ± 3.3	98.7 ± 3.3	97.6 ± 3.2	96.0 ± 3.2	100

**Table 2 pharmaceutics-14-01249-t002:** The iron load quantification MFNP internalized into hBM-MSC by the NIRF signal intensity using different labeling conditions.

iT (h)	NIRF Intensity (Photons/s) × 10^8^								
		Without MF	With MF		Without MF	With MF		Without MF	With MF
3	[10 µg Fe/mL] in cell labeling	0.35 ± 0.02	2.21 ± 0.08	[30 µg Fe/mL] in cell labeling	1.04 ± 0.05	6.96 ± 0.35	[50 µg Fe/mL] in cell labeling	1.80 ± 0.13	9.57 ± 0.48
6		1.21 ± 0.06	3.98 ± 0.17		2.78 ± 0.14	13.05 ± 0.65		3.53 ± 0.19	15.66 ± 0.78
8		1.47 ± 0.08	4.81 ± 0.23		3.48 ± 0.17	15.66 ± 0.78		4.30 ± 0.23	17.40 ± 0.87
12		1.57 ± 0.08	6.09 ± 0.30		3.92 ± 0.20	18.27 ± 0.91		5.74 ± 0.29	23.36 ± 1.22
18		1.74 ± 0.09	7.66 ± 0.38		4.35 ± 0.22	21.75 ± 1.09		6.26 ± 0.31	24.80 ± 1.24
24		2.00 ± 0.10	8.09 ± 0.39		4.44 ± 0.22	22.62 ± 1.13		6.96 ± 0.35	25.23 ± 1.26

Abbreviations: NIRF—near-infrared fluorescence image; iT—incubation time; MF—magnetic field.

**Table 3 pharmaceutics-14-01249-t003:** The iron load quantification MFNP internalized into hBM-MSC by the ICP-MS signal intensity using different labeling conditions.

iT (h)	* Iron Load/Cell (*p*g Fe/Cell) Number MFNP per Cell × 10^4^								
		Without MF	With MF		Without MF	With MF		Without MF	With MF
3	[Fe] in cell labeling = 10 µg Fe/mL	* 0.64 ± 0.04	* 2.21 ± 0.15	[Fe] in cell labeling = 30 µg Fe/mL	* 1.40 ± 0.10	* 6.96 ± 0.49	[Fe] in cell labeling = 50 µg Fe/mL	* 3.34 ± 0.23	* 13.50 ± 0.94
		0.81 ± 0.04	2.78 ± 0.14		1.76 ± 0.09	8.74 ± 0.44		4.20 ± 0.21	16.96 ± 0.85
6		* 1.29 ± 0.09	* 4.40 ± 0.31		* 3.40 ± 0.24	* 16.36 ± 1.14		* 4.74 ± 0.33	* 21.49 ± 1.50
		1.62 ± 0.08	5.53 ± 0.28		4.27 ± 0.21	20.55 ± 1.03		5.96 ± 0.30	26.99 ± 1.35
8		* 1.53 ± 0.11	* 5.71 ± 0.40		* 3.94 ± 0.28	* 18.87 ± 1.32		* 5.49 ± 0.38	* 23.69 ± 1.66
		1.92 ± 0.10	7.18 ± 0.36		4.95 ± 0.25	23.71 ± 1.19		6.89 ± 0.34	29.76 ± 1.49
12		* 1.74 ± 0.12	* 7.29 ± 0.51		* 4.73 ± 0.33	* 21.53 ± 1.51		* 6.57 ± 0.46	* 28.16 ± 1.97
		2.19 ± 0.11	9.15 ± 0.46		5.94 ± 0.30	27.05 ± 1.35		8.26 ± 0.41	35.38 ± 1.77
18		* 1.96 ± 0.14	* 8.71 ± 0.61		* 5.34 ± 0.37	* 24.10 ± 1.69		* 7.43 ± 0.52	* 29.29 ± 2.05
		2.46 ± 0.12	10.95 ± 0.55		6.71 ± 0.34	30.28 ± 1.51		9.33 ± 0.47	36.79 ± 1.84
24		* 2.16 ± 0.15	* 9.31 ± 0.65		* 5.56 ± 0.39	* 25.09 ± 1.76		* 8.00 ± 0.56	* 29.17 ± 2.04
		2.71 ± 0.14	11.70 ± 0.59		6.98 ± 0.35	31.52 ± 1.58		10.05 ± 0.50	36.65 ± 1.83

Note: * represents the value of iron load per cell. Abbreviations—MFNP: multifuctional nanoparticle; MF: magnetic field; iT: incubation time.

**Table 4 pharmaceutics-14-01249-t004:** The maximum concentrations of nanoparticles ($C_{MFNP}^{max}$) and the characteristic time of internalization (*τ*_*MFNP*_) according to Equation (1), and the recommended incubation time (t_90%_C24 h_) to reach 90% of the concentration of MFNP internalized in 24 h (90% C24 h).

[Fe] In hBM-MSC Labeling (µg Fe/mL)	$C_{MFNP}^{max}$ (*p*g Fe/cell)	$\tau_{MFNP}$ (h)	t_90%_C24 h_ (h)	90% C24 h
(10)	2.50 ± 0.15	5.51 ± 0.86	12.8	1.85
(30)	7.14 ± 0.32	4.93 ± 0.51	12.4	4.91
(50)	7.56 ± 0.68	4.42 ± 0.33	15.6	7.17
(10) + Magnetic Field	11.10 ± 0.19	8.83 ± 0.54	16.7	8.44
(30) + Magnetic Field	31.43 ± 2.30	4.58 ± 0.53	11.2	21.98
(50) + Magnetic Field	35.24 ± 0.15	4.28 ± 0.52	10.5	26.60

Abbreviations—[Fe]: iron concentration; hBM-MSC: human bone marrow mesenchymal stem cell; MFNP: multifunctional nanoparticle.

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
