# Peer review of "Optimization of Multimodal Nanoparticles Internalization Process in Mesenchymal Stem Cells for Cell Therapy Studies"

_pharmaceutics, 2022, doi:10.3390/pharmaceutics14061249_

Round 1

Reviewer 1 Report

Q1: Please describe how MFNPs were functionalized with fluorescent dyes.  Is it covalent binding and what kind of dyes were attached?

Line 15: “ Considering the difficulties and limitations of labeling stem cells using multifunctional 15 nanoparticles (MFNP) and monitoring these labeled stem cells. The purpose of this study…

I guess it should be comma instead of dot “…stem cells, the purpose…”. Otherwise it is uncompleted sentence.

Line 29: “These optimized labeling parameters promoted an increase NIRF signal of 215% 29 at 1h and 201% at 7h with good BLI signal over time, without significant differences between unlabeled or labeled cells.

Does that mean that unlabeled cells showed the same signal increase? Which parameter did not change for unlabeled or labeled cells

Line 356: “Technical Information 356 of this nanoparticle (35 nm)” Description of MFNP is missing. Are they commercially available or were synthetized by a protocol?

Line 20:  50 ug Fe/mL – is it any kind of SI unit?

Line 17 and 119:”mwsenchymal cells” – a typo

Line 473: “de” should be replaced with “from”

Line 474: How can you explain that 4 fold BLI signal growth corresponds to 13-fold increase of internalized MFNP?  Usually fluorescence intensity is linearly increases with concentration.

Line 528: (E) graph is not described

Line 549: “modification” should be replaced with “modified”

Author Response

Reviewer #1

  • Please describe how MFNPs were functionalized with fluorescent dyes. Is it covalent binding and what kind of dyes were attached?

Answer: Thank you for your question. As described in item 2.4 “Multifunctional nanoparticles with magnetic and fluorescent properties”, the product is commercial and was purchased from Biopal.

  • Line 15: “ Considering the difficulties and limitations of labeling stem cells using multifunctional 15 nanoparticles (MFNP) and monitoring these labeled stem cells. The purpose of this study…”? I guess it should be comma instead of dot “…stem cells, the purpose…”. Otherwise it is uncompleted sentence.

Answer: Thank you for your observation. We corrected the phase appointed in the manuscript.

  • Line 29: “These optimized labeling parameters promoted an increase NIRF signal of 215% at 1h and 201% at 7h with good BLI signal over time, without significant differences between unlabeled or labeled cells.” Does that mean that unlabeled cells showed the same signal increase? Which parameter did not change for unlabeled or labeled cells

Answer: Thanks for your question. The sentence was rewritten, improving the information regarding the result of the signal strength of the NIRF and BLI in the in vivo experiment. In NIRF only labeled cells showed increased signal (cells labeled with a magnetic field in relation to cells labeled without a magnetic field). In the case of the BLI signal, which depends on the number of cells and not on the cellular labeling, the influence of the labeling condition did not interfere with the BLI signal.

  • Line 356: “Technical Information of this nanoparticle (35 nm)” Description of MFNP is missing. Are they commercially available or were synthetized by a protocol?

Answer: Thank you for your question. This nanoparticle is commercially available by the Biopal company (Molday ION™, Worcester, MA, USA) as described in item 2.4 of the method section of the manuscript.

  • Line 20: 50 ug Fe/mL – is it any kind of SI unit?

Answer: Thank you for your observation. We corrected the error in the manuscript

  • Line 17 and 119:”mwsenchymal cells” – a typo?

Answer: Thank you for your observation. We corrected the error in the manuscript

  • Line 473: “de” should be replaced with “from”?

Answer: Thank you for your observation. We corrected the error in the manuscript.

  • Line 474: How can you explain that 4 fold BLI signal growth corresponds to 13-fold increase of internalized MFNP? Usually fluorescence intensity is linearly increases with concentration.

Answer: Thank you for your observation. Because of a calculation error, the text has been changed from "4 times" to "13 times." The same relationship as in the NIRF signal was included as well. Regarding the fluorescence intensity, depending on the range, it has a linear behavior, at high concentrations; it does not present a linear behavior.

  • Line 528: (E) graph is not described

Answer: Thank you for your observation. We added the description of item (E) in Figure 5.

  • Line 549: “modification” should be replaced with “modified”

Answer: Thank you for your observation. We corrected the error in the manuscript

Reviewer 2 Report

Authors in the manuscript have performed an interesting study, the manuscript can be accepted after addressing the comments below.

Abstract: First sentence has to be revised, it is incomplete.

Line 25-36: 4 times higher to what?

Abstract has to be reworded and modified. It contains long sentences and confusing language was used.

Introduction:

Please provide full form of MSCs

Authors have discussed the disadvantages of the available techniques. However, authors need to discuss the challenges and limitations of each method and how the proposed method in the investigation offers benefit.

Methods: Better to use the word “informed consent was received from all participants in the trial”

The mononuclear cells were obtained from bone marrow of 122 healthy donors.

cells were centrifuged without brake for 30 min. What does brake mean?

What is de CO2?

Quantification of MFNP internalized into hBM-MSC using NIRF and ICP-MS, and the 246 correlation between them. Please delete after ICP-MS

The language of the manuscript has to be improved a lot. Please avoid long sentences and clearly communicate the findings of the study.

The paper looks too long. Please check if some sections can be summarized and provided as a supplementary to this paper.

Please provide conclusions for the study.

Author Response

Reviewer #2

  • Abstract: First sentence has to be revised, it is incomplete.

Answer: Thank you for your observation. We corrected this sentence in the manuscript

  • Abstract: Line 25-36: 4 times higher to what?

Answer: Thank you for your observation. We corrected the sentence of the manuscript

  • Abstract has to be reworded and modified. It contains long sentences and confusing language was used.

Answer: We have improved the abstract and the general language, and reduced the long sentences.

  • Introduction: Please provide full form of MSCs

Answer: Thank you for the observation, we have provide the full form of MSCs (mesenchymal stem cells) 

  • Introduction: Authors have discussed the disadvantages of the available techniques. However, authors need to discuss the challenges and limitations of each method and how the proposed method in the investigation offers benefit

Answer: Thank you for your suggestion. We added complementary information about the studies [1-5] that reported the advantages and disadvantages of each method.

  • Methods: Better to use the word “informed consent was received from all participants in the trial”

Answer:  We modified and added the information about the informed consent.

  • Methods: The mononuclear cells were obtained from bone marrow of 122 healthy donors.

Answer: Thank you for your observation. We have added complementary data about the number of healthy donors used in this study. The question has mistakenly included the number 122 which only refers to the line number for this sentence in the manuscript.

  • Methods: cells were centrifuged without brake for 30 min. What does brake mean?

Answer: When the centrifugation is set without brake, the centrifuge stop by just running down till the rotor stops by itself, not disturbing the solution by an abrupt stop.

  • Methods: What is de CO2?

Answer: Sorry, we corrected this phrase.

  • Methods: Quantification of MFNP internalized into hBM-MSC using NIRF and ICP-MS, and the 246 correlation between them. Please delete after ICP-MS

Answer: Thank you for your suggestion. We corrected the subtitle of the manuscript

The question has mistakenly included the number 246 which only refers to the line number for this sentence in the manuscript.

  • The language of the manuscript has to be improved a lot. Please avoid long sentences and clearly communicate the findings of the study.

Answer: Thank you for your observation and suggestion. We have reviewed the manuscript's long sentences and increased the clarity and explanation of the study findings.

  • The paper looks too long. Please check if some sections can be summarized and provided as a supplementary to this paper.

Answer: Thank you for your observation. We have shortened some parts of the manuscript, summarizing the information.

  • Please provide conclusions for the study.

Answer: We have added a conclusion section in the manuscript.

Reference

  1. Neri, M.; Maderna, C.; Cavazzin, C.; Deidda-Vigoriti, V.; Politi, L.S.; Scotti, G.; Marzola, P.; Sbarbati, A.; Vescovi, A.L.; Gritti, A. Efficient In Vitro Labeling of Human Neural Precursor Cells with Superparamagnetic Iron Oxide Particles: Relevance for In Vivo Cell Tracking. Stem Cells 2007, 26, 505-516, doi:10.1634/stemcells.2007-0251.
  2. Landázuri, N.; Tong, S.; Suo, J.; Joseph, G.; Weiss, D.; Sutcliffe, D.J.; Giddens, D.P.; Bao, G.; Taylor, W.R. Magnetic targeting of human mesenchymal stem cells with internalized superparamagnetic iron oxide nanoparticles. Small 2013, 9, 4017-4026, doi:10.1002/smll.201300570.
  3. Mamani, J.B.; Souza, T.K.F.; Nucci, M.P.; Oliveira, F.A.; Nucci, L.P.; Alves, A.H.; Rego, G.N.A.; Marti, L.; Gamarra, L.F. In Vitro Evaluation of Hyperthermia Magnetic Technique Indicating the Best Strategy for Internalization of Magnetic Nanoparticles Applied in Glioblastoma Tumor Cells. Pharmaceutics 2021, 13, 1219.
  4. Pongrac, I.M.; Radmilović, M.D.; Ahmed, L.B.; Mlinarić, H.; Regul, J.; Škokić, S.; Babič, M.; Horák, D.; Hoehn, M.; Gajović, S. D-mannose-Coating of Maghemite Nanoparticles Improved Labeling of Neural Stem Cells and Allowed Their Visualization by ex vivo MRI after Transplantation in the Mouse Brain. Cell Transplantation 2019, 28, 553-567, doi:10.1177/0963689719834304.
  5. Jasmin; Torres, A.L.; Jelicks, L.; de Carvalho, A.C.; Spray, D.C.; Mendez-Otero, R. Labeling stem cells with superparamagnetic iron oxide nanoparticles: analysis of the labeling efficacy by microscopy and magnetic resonance imaging. Methods Mol Biol 2012, 906, 239-252, doi:10.1007/978-1-61779-953-2_18.

Reviewer 3 Report

Dear authors,

The authors reported the direct labeling of mesenchymal stem cells (MSC), using or not magnetic field to introduce multifunctional particle, including iron, and conjugated with 2 fluorophores. After determining the optimal conditions for MSC labeling (higher concentration combined with the magnetic impulse), the authors proved the MSC to be viable, with very low apoptosis (les than 4%). Labeled MSC were injected in the brain, and migrated into the ischemic area. MSC were present for 7 h at least, in the rat brain. Unfortunately, there is no data bout the effect of MSC on the ischemia. Labeling cells, for cell/gene therapy is a major field of study, that require a lot test and studies to allow long term follow up after transplantation. 

I have some questions about the manuscript:

- Some editing over the manuscript is required: e.g line 27

- What is the manufacturer or how the authors obtained the lentivirus?

- Did the authors study the decrease of the MSC labeling, during the MSC proliferation (in cell culture dish or in after brain injection), because the signal should be divided by 2 at least, each time MSC are dividing?

- Did the authors follow up the animal for longer time to observe a decline in the signaling and also to study the biodistribution of the MSC over the rat body? These information will be very helpful.

- The goal of injected MSC is also to restore the local blood perfusion. Did the authors measure the local blood perfusion after MSC are injected?

- If this methodology is used for human, do the authors think it's possible to use it for patients?

- in the discussion, the authors mentioned that FBS is important for cell labeling, using such particles. However, FDA has major concerns about FBS that could contain bovine spongiform encephalopathy (BSE). There is a risk with this method to inject directly the BSE in the brain of the patients. can the authors discuss about alternative methodologies avoiding to use FBS, to label the MSC?

Author Response

Reviewer #3

The authors reported the direct labeling of mesenchymal stem cells (MSC), using or not magnetic field to introduce multifunctional particle, including iron, and conjugated with 2 fluorophores. After determining the optimal conditions for MSC labeling (higher concentration combined with the magnetic impulse), the authors proved the MSC to be viable, with very low apoptosis (les than 4%). Labeled MSC were injected in the brain, and migrated into the ischemic area. MSC were present for 7 h at least, in the rat brain. Unfortunately, there is no data about the effect of MSC on the ischemia. Labeling cells, for cell/gene therapy is a major field of study, that require a lot test and studies to allow long term follow up after transplantation. 

I have some questions about the manuscript:

  • Some editing over the manuscript is required: e.g line 27

Answer: Thank you for your observation. We have edited line 27 and verified the spelling over the manuscript.

  • What is the manufacturer or how the authors obtained the lentivirus?

Answer: Thank you for your question. The lentiviral vector pMSCV Luc2 T2A Puro was provided by Dr. Deivid de Carvalho Rodrigues as described in previous studies [1-3] and these references were included in the manuscript.

  • Did the authors study the decrease of the MSC labeling, during the MSC proliferation (in cell culture dish or in after brain injection), because the signal should be divided by 2 at least, each time MSC are dividing?

Answer: The decay of the fluorescence signal is based, among other factors, in the number of labeled cells, process of nanoparticles exocytosis the cells and in the detection sensitivity of the contrast agent by the fluorescence technique, in this sense, if the cells are closer, it will be possible to capture the fluorescence signal, but if these cells are very spread out, the signal detection will be lower, due to the loss in sensitivity by the technique.

In this study, during the short period of evaluation in vivo, the fluorescence signal remained almost constant, because the load of iron internalized in the cells displayed just small variations independent of their proliferation (acute stage). Exocytosis process may have contributed to a discrete decrease and non-significant in signal, as shown in Figure 6.

The objective of this study was to find the best cell labeling condition for the emission of higher fluorescence signal in vivo (acute stage), however, we believe that it would be interesting for future studies to evaluate the signal for a longer periods.

Thus, in longer periods of observation in vitro and in vivo, it would be possible to understand the behavior of the signal intensity over time and verify the influence of cell division/proliferation on the signal detection process.

  • Did the authors follow up the animal for longer time to observe a decline in the signaling and also to study the biodistribution of the MSC over the rat body? These information will be very helpful.

Answer: No, the proposal of the study was to evaluate only the cellular homing and tracking in the acute stage, as also the difference in NIRF intensity signal according to nanoparticle internalization with or without a magnetic field.

However, the BLI and NIRF techniques allowed evaluating cell biodistribution over the rat body in this acute stage and the signal was detected in the animal's cranium, as shown in figure 6. Regarding the evaluation of cell biodistribution for longer periods, it was mention among the study limitations in a paragraph in the discussion section. 

  • The goal of injected MSC is also to restore the local blood perfusion. Did the authors measure the local blood perfusion after MSC are injected?

Answer: Thank you for your suggestion. The local blood perfusion analysis was performed only to confirm the decrease in perfusion after phototromboses induction, comparing the local perfusion before and after ischemic induction. Unfortunately, we did not perform the blood perfusion analysis surrounding the ischemic lesion after hBM-MSC injection; however, it is an interesting proposal for future studies.

  • If this methodology is used for human, do the authors think it's possible to use it for patients?

Answer: Thank you for your question. The methodology could be used for clinical application, allowing the assessment of cells homing and tracking by magnetic resonance image due to magnetic characteristics of the multimodal nanoparticles adopted here.  However, for that, would be necessary adaptations in the cells manufacture, as the use of cGMP reagents, removal of antibiotics and animal products from production, minimizing the risk for patients.

  • In the discussion, the authors mentioned that FBS is important for cell labeling, using such particles. However, FDA has major concerns about FBS that could contain bovine spongiform encephalopathy (BSE). There is a risk with this method to inject directly the BSE in the brain of the patients. Can the authors discuss about alternative methodologies avoiding to use FBS, to label the MSC?

Answer: Thank you for your question. For clinical applications the FBS can be replaced by human pooled AB serum. Human serum is commercially available and can be obtained with good practice manufacture certificate (cGMP) that is required for clinical use. Herein we have used FBS due to the lower cost and because this study still in pre-clinical phase. We have included in the discussion this option for the more advanced translational phases of this study.

Reference

  1. Souza, T.K.F.; Nucci, M.P.; Mamani, J.B.; da Silva, H.R.; Fantacini, D.M.C.; de Souza, L.E.B.; Picanço-Castro, V.; Covas, D.T.; Vidoto, E.L.; Tannús, A.; et al. Image and motor behavior for monitoring tumor growth in C6 glioma model. PloS one 2018, 13, e0201453, doi:10.1371/journal.pone.0201453.
  2. Kuroda, H.; Kutner, R.H.; Bazan, N.G.; Reiser, J. Simplified lentivirus vector production in protein-free media using polyethylenimine-mediated transfection. Journal of Virological Methods 2009, 157, 113-121, doi:https://doi.org/10.1016/j.jviromet.2008.11.021.
  3. da Silva, H.R.; Mamani, J.B.; Nucci, M.P.; Nucci, L.P.; Kondo, A.T.; Fantacini, D.M.C.; de Souza, L.E.B.; Picanço-Castro, V.; Covas, D.T.; Kutner, J.M.; et al. Triple-modal imaging of stem-cells labeled with multimodal nanoparticles, applied in a stroke model. World J Stem Cells 2019, 11, 100-123, doi:10.4252/wjsc.v11.i2.100.

Round 2

Reviewer 2 Report

Thanks for addressing the comments. 

Please remove "Therefore" from the begining of conclusion paragraph.

Line 4 of conclusion, please change significant to significantly

Reviewer 3 Report

Dear Authors,

I have no additional comments.

Sincerely